# “Who Takes Care of Carers?”: Experiences of Intensive Care Unit Nurses in the Acute Phase of the COVID-19 Pandemic

**DOI:** 10.3390/healthcare12020162

**Published:** 2024-01-10

**Authors:** Marina Castaño-García, José Granero-Molina, Alba Fernández-Férez, Isabel María Fernández-Medina, María Isabel Ventura-Miranda, María del Mar Jiménez-Lasserrotte

**Affiliations:** 1Andalusian Health Service, 04009 Almería, Spain; marina.castano.sspa@juntadeandalucia.es (M.C.-G.); alba.fernandez.ferez.sspa@juntadeandalucia.es (A.F.-F.); 2Nursing, Physiotheraphy and Medicine Department, University of Almería, 04120 Almería, Spain; isabel_medina@ual.es (I.M.F.-M.); mvm737@ual.es (M.I.V.-M.); mjl095@ual.es (M.d.M.J.-L.); 3Faculty of Health Sciences, Universidad Autónoma de Chile, Santiago 7500000, Chile

**Keywords:** qualitative study, intensive care unit, SARS-CoV-19, COVID-19 pandemic, nurses

## Abstract

**Introduction:** The COVID-19 pandemic caused an international health emergency situation where nursing took on a fundamental role. The high number of patients in hospital ICUs led to a shift in nurses’ working conditions and workload. **Objective:** The objective of this study was to describe the experiences of nurses who worked in ICUs during the acute phase of the COVID-19 pandemic. **Methodology:** A qualitative, descriptive study was carried out, with the participation of 21 nurses who worked in the ICU during the pandemic. Data collection took place between May and July 2021 through 21 in-depth interviews. **Results:** Three main themes emerged: (1) COVID-19 in ICUs: nurses on the frontline. (2) United against adversity: teamwork. (3) New optics of critical care and the nursing profession. COVID-19 was perceived with harshness, and the lack of knowledge about the virus generated confusion, anxiety and fear due to the risk of transmission to family members and relatives. The pandemic marked a shift in the management of human, material and economic resources. Novice nurses learned critical care at an accelerated pace, with significant physical and psychological strain. Expert nurses carried the burden of training new nurses. Although there were tense situations, experiencing these adverse situations as a team led to feelings of increased belonging, togetherness and professional bonding for nurses. While the participants noted an increase in motivation to continue in their profession, they also had a feeling of not having been cared for as they deserve by healthcare institutions.

## 1. Introduction

At the end of 2019, the World Health Organisation (WHO) defined the emerging virus in the city of Wuhan (China) as 2019-nCoV. This virus, pandemic and global health emergency spread rapidly, and became known as COVID-19 [1]. According to data from the WHO, from the onset of the pandemic until November 2023, there were 771,820,937 cases of COVID-19 reported worldwide, with 6,978,175 deaths. In Spain, from January 2020 to November 2023, 13,980,340 cases of COVID-19 were reported, with 121,852 deaths [2]. Given the severity and high morbidity and mortality rates of the disease, 48,709 patients in Spain required admission to ICUs (Intensive Care Units). In Andalusia, a large region in southern Spain, 90,076 patients were admitted to hospital for COVID-19, 7407 of them to the ICU [3], resulting in 2417 deaths [3]. The acute phase of the COVID-19 pandemic in Spain started in February 2020, with the appearance of the first case, until March 2022, with the establishment of the new COVID-19 Surveillance and Control Strategy [4]. The sixth wave of the COVID-19 pandemic marked a transition to a strategy of surveillance, control and protection of the most vulnerable individuals/areas and severe cases. In Spain’s case, the COVID-19 occupancy rate has remained stable in all age groups since March 2023 [3], ranging from 1 to 6% [5].

The COVID-19 pandemic led to restructuring of hospital spaces and human and material resources, which especially affected ICUs and nurses [6]. The ICU in our study was originally equipped with 28 beds for patients with coronary pathologies, polytrauma and other serious diseases, but was fully occupied by COVID-19 patients at the height of the pandemic (COVID-ICU). The remaining patients were sent to post-operatory and resuscitation units. Faced with an overwhelming number of new cases, an additional 40 beds were added in January 2021, converting a Hospital Care Unit into an ICU [7]. This was a major change, as quality care, infection control, patient safety and staff well-being depend, to a large extent, on the clinical and structural environments where care is administered [8].

In the COVID-ICU, ICU nurses worked alongside surgery nurses, resuscitation nurses and hospital intake nurses. The experience of caring for critically ill patients is related to safety, clinical effectiveness and improved outcomes [9,10]. The nursing care of a critically ill patient encompasses general and specific care, attention to basic needs, and advanced practice in a highly technological environment [11,12]. ICU nurses devote time to patient care, as well as clinical responsibilities, interaction with family members, and participation in decision making [13,14]. An ICU nurse’s workload is both physical and psychological, which influences the quality of the care they provide [15,16]. Heavy ICU workload during the COVID-19 pandemic may have affected patient safety, incidence of errors, nosocomial infections, and morbidity and mortality [17]. High numbers of patient admissions in the COVID-ICU increased the workload of the nurses there, who may have also been responsible for supervising inexperienced nurses [18]. The use of personal protective equipment (PPE), not leaving the ICU during their shifts, nor having breaks during shifts, led to work overload [18,19]. The COVID-19 pandemic brought on a new “status quo” in the functioning and safety of critical patient care [20], but also for nurses [21]. While the impact of the COVID-19 pandemic in ICUs has been studied from an epidemiological, diagnostic, clinical or treatment perspective [22,23,24], there is a need to explore the experiences of nurses, along with the repercussions of COVID-19 on the concept of the profession and critical care nursing. While some qualitative research has addressed the issue [25,26,27,28], information about changes in the perception of care in nurses who worked on the front line during the acute phase of the pandemic is scarce. Our study asks the questions: What were the experiences of ICU nurses like in the acute phase of the COVID-19 pandemic? How did the COVID-19 pandemic impact care, patient safety, and the concept of the critical care nursing profession? The aim of our study is to describe and understand ICU nurses’ experiences of caring for critically ill patients in the acute phase of the COVID-19 pandemic.

## 2. Materials and Method

### 2.1. Design

A qualitative descriptive study was carried out. This methodology allows us to describe phenomena through the perceptions or experiences of the participants themselves in their natural state, interpreting their views of the situations experienced [29]. This approach is suitable for exploring and understanding the experiences of nurses who cared for ICU patients during the acute phase of the COVID-19 pandemic [30]. In writing the manuscript, the Consolidated Criteria for Reporting Qualitative Research were applied (COREQ) [31].

### 2.2. Sample

All participants were selected by purposive sampling, and a total of 21 nurses were recruited. Inclusion criteria were as follows: having a nursing degree, being between 18 and 60 years of age, having worked in ICUs during the acute phase of the COVID-19 pandemic (February 2020 to March 2022), at least 6 months of ICU experience, and consent to participate in the study. Exclusion criteria were as follows: part-time participants working ≤60% of the full working day and refusing to participate. The study was carried out in a university hospital in southern Spain between 2021 and 2022. Sociodemographic characteristics of the participants are shown in Table 1. A total of twenty-six ICU nurses agreed to participate, four eventually declined to participate due to lack of time for the interview, and one eventually withdrew due to illness. One of the researchers was an ICU nurse during the acute phase of the pandemic, facilitating the participant recruitment process.

### 2.3. Data Collection

After obtaining permission from the Ethics and Research Committee (EFM 187/2022), the researchers contacted the participants. Data collection took place in a classroom provided by the University in May 2022. In-depth individual interviews were performed to allow for the nurses’ narratives (see Table 2). All interviews were carried out in the Spanish language. Participants were interviewed only once, and interviews were audio-recorded and transcribed verbatim. Data collection ceased when data saturation was reached. The interviews lasted an average of 48 min. Researchers who conducted the interviews (A.F.F. and M.C.G.) were nurses with extensive ICU experience. Prior to the interviews, the researchers informed the participants about the objective of the study, their anonymity, and collected their sociodemographic data and informed consent.

### 2.4. Data Analysis

After the interviews were transcribed, they were incorporated into a hermeneutic unit and analysed using ATLAS.ti 22 software. Data analysis was based on the steps of the thematic analysis described by Braun and Clark, 2006 [32]. Phase 1: comprehensive reading of the interview transcripts to get a general overview; subsequent rereading to familiarise oneself with the data. Phase 2: semantic or latent content codes were systematically generated. Phase 3: after coding the interviews, the codes were analysed by inductively generating sub-themes. Phase 4: analysis of emerging themes, avoiding redundancy, reviewing their consistency and ensuring they are supported by codes. Phase 5: everything involved in each theme was detailed and defined. Phase 6: the report was written and includes the most representative fragments of the interviews, relating them to the questions and themes.

### 2.5. Rigour

To ensure the rigour of the study, the following quality criteria were used [33]. Credibility: the data collection process was detailed and the interpretation of the data was supported by the investigators; and the analytical process was reviewed by two independent reviewers. Transferability: detailed description of study participants, context and method. The research team triangulated the data analysis: two researchers performed the initial data analysis; themes and subthemes were discussed with main researchers, removing those that did not reach at least 66% agreement. Reliability: two experts, with experience in the ICU, outside the data collection and analysis, examined the interpretation verifying a final list of themes and subthemes. Confirmability: all researchers read the transcripts independently, reaching agreement on units of meaning, themes and subthemes. Some participants had the opportunity to view the transcripts. The research team triangulated the data analysis.

### 2.6. Reflexivity

The research team members worked in the ICU during the COVID-19 pandemic, which modelled their expectations, the power dynamic between participants and researchers, and minimised disagreements. The methodology supports the stance of the researchers, who see the need to rethink the profession in general, the social support received by nurses, and the opinion of those involved in critical care during the COVID-19 pandemic. The research was carried out in an unprecedented historical time, in which the professional and personal expectations of nurses were put into continuous question [34].

### 2.7. Ethical Considerations

The study was approved by the University of Almería Ethics and Research Committee of the Department of Nursing Physiotherapy and Medicine (EFM 187/2022). It was carried out following the guidelines of the Declaration of Helsinki. Participants were informed about the purpose of the study prior to the interviews, and that the data would be handled confidentially and for research purposes only. Participants gave informed consent to participate in the study.

## 3. Results

A total of 21 ICU nurses were interviewed (Table 3). The mean age was 34.04 years, SD = 7.4. Regarding sex, 71.4% were female and 28.6% male. Regarding marital status, 33.3% were married, 4.7% divorced, and 62% single. In total, 62% of the nurses did not have children. The average length of the nurses’ professional experience was 9.4 years, with 4.2 average years of experience in ICUs. Three main themes and eighth sub-themes were extracted from inductive data analysis.

### 3.1. THEME 1: COVID-19 in ICUs: Nurses on the Frontline

This theme encompasses the feelings and concerns that surfaced in nurses with the onset of the pandemic. The acute phase of the COVID-19 epidemic was perceived as entering a “battlefield”. Nurses developed feelings of fear and ethical dilemmas; but they also felt the importance of humanising care, accompanying the patient and dignifying the end of life.

Nursing on the battlefield: in the trenches.

Participants shared the perception of being on a battlefield, fighting against a relentless and unknown opponent that was cornering them. The feeling of chaos, lack of control and unfamiliarity generated high levels of anxiety before going to work and throughout their shifts. As one nurse expresses:


*“New patients never stopped coming in, it was out of control... You saw people coming in and shutting down, their bodies could not cope with the virus....you knew that many people were left waiting for a respirator, and that many others were dying waiting for it...”*
[IDI7]

The lack of information about the virus raised concerns in nurses about how to protect themselves effectively, and about the possibility of infecting their family members.


*“At the beginning it was not clear how to protect yourself against the disease. This generated insecurity and fear among professionals (...) Fear of the unknown, of contagion… more than fear of me getting infected, of my family getting infected.”*
[IDI5]

When a family member is especially vulnerable to infection, the situation is exacerbated and ethical and moral dilemmas arise for ICU nurses. Some of them had very vulnerable family members and did not know what to do.


*“My husband (a kidney transplant recipient) wanted me to resign for fear of infecting him (...) That night I didn’t go to work, and in the following days I had time to become aware that this was what I had to do, and I would have to get through it any way I could (...)”*
[IDI6]

Questioning the quality of care.

Nurses noted that the quality of care for patients suffering from COVID-19 was compromised. Factors such as unfamiliarity with critical care generated frustration. The lack of more time to care for each patient (occupational overload), together with the deficit of experience and safety in the development of procedures, generated tension in many nurses.


*“Lack of time, lack of resources, lack of privacy for patients, lack of autonomy (...) Yes, there were certainly deficits in care, without a doubt.”*
[IDI3]

In contrast, other participants emphasise that the quality of care was maintained at all times, and increased as the disease became more widely known.


*“There was no lack of material, no lack of means. There was only a lack of information about the virus and the disease. It couldn’t be any different because no one knew its effects, but as the months went by, care and treatment became more and more successful.”*
[IDI15]

Humanising care and providing company: death in isolation.

The need for the humanisation of critical care was a major challenge during the pandemic. Nurses showed greater sensitivity, observing how patients suffered alone. The loneliness and isolation to which many patients were subjected to at the time of their death is perceived by nurses as a fundamental shortcoming. Participants alluded to the fact that no one should die alone and noted that being accompanied was part of having dignity at the end of life.


*“Death in isolation for me was the worst, holding hands with patients as they stopped breathing and died. The treatment of the body of the deceased, towards their families, who waited without knowing …, sometimes there was a lack of attention to dignity.”*
[IDI4]

Some participants had the opportunity to accompany a family member who had COVID. It was at that moment when the nurses truly became aware of the gravity of the situation.


*“Unfortunately, I experienced the death of one of my own family members in the COVID area. Being a worker at the hospital, I was given the option to stay with her wearing my PPE. Thanks to that, I was able to accompany her in her last days and in her final moments, … at that point, I was fully aware of the loneliness in which the rest of the patients were dying (…) God only knows the people who died in this kind of painful isolation, the number of relatives who couldn’t say goodbye and accompany their loved ones, where is the dignity in these deaths?”*
[IDI12]

### 3.2. THEME 2: United against Adversity: Teamwork

This topic addresses the nurses’ experiences caring for critically ill patients in the acute phase of the pandemic. Novice nurses had to learn against the clock, which added extra responsibility for the experienced nurses. The pressure of providing care to all and the lack of knowledge of the disease generated doubts about the safety of the professionals and led to conflicting feelings within the team.

Observing experienced nurses and learning by doing.

Some participants were relocated to the ICU due to an increased need for staff. The new nurses were forced to learn against the clock, which generated negative experiences linked to stress. More veteran nurses managed the situation because of their skills, competence and years of experience. Faced with the intense pressure of caregiving, they doubled their efforts as patient caregivers and mentors to the new nurses.


*“Learning to work in the ICU with foggy glasses, 3 pairs of gloves and unbearable heat (clothing). A patient would come in and you had to do a central catheterization to administer medication urgently, an artery to check their respiratory status and blood pressure, an intubation because they came in with their mouth open, like a fish out of water. In those situations, you cannot stop to teach the newcomers, they just learn by watching the veteran nurses.”*
[IDI10]


*“The veteran nurses were overloaded with their patients and training the newcomers, they were burnt out from teaching so much. Sometimes it was hard to ask them for favours when we had to perform a technique that we did not know how to do, but they always helped us to do it so that we could learn, or they did it themselves, … we will be eternally grateful for that.”*
[IDI13]

When safety starts to falter.

The participants highlighted weaknesses in professional safety. The lack of knowledge surrounding the use of PPE and the amount of time spent in the isolation rooms, led to physical and psychological fatigue. Some of the nurses saw colleagues’ sheer exhaustion after caring for a patient for hours wearing PPE in isolation rooms. Regarding patient safety, the priority was always to save their lives.


*“In my opinion, it was never diminished (safety). We always knew what the priority was, and it was them [the patients]. In an emergency, there was hardly time to put on PPE, but if it was necessary to save their life, we went into the [isolation] room.”*
[IDI11]

Coming together to withstand the pandemic.

The nurses experienced moments of extreme tension in the ICU. Stress, fear and tension brought about feelings of selfishness and strained relationships among team members. However, participants emphasised that these negative feelings were eventually resolved and did not overshadow their sense of teamwork. Feeling that they were all in the same situation strengthened bonds, and nurses felt united through adversity. These situations created deep bonds between the nurses who worked together in ICU in the acute phase of the COVID-19 pandemic. There were many different feelings, both positive and negative:


*“We did not have many NBC masks, … they had been donated by farmers in the province. Some people kept them in their lockers at the end of their shift (they didn’t care if their colleagues needed them). I also gained a lot of good things out of the experience, but the best without any doubt is the people I worked with, great colleagues and professionals, and more than that, friends.”*
[IDI19]


*“The bond between nurses grew, there was much more trust. There was no more tension between us, but a lot of support, respect and understanding (…) If I talk about the new colleagues, it’s the same, because everything was new for everyone. I welcomed them with open arms and I still have a friendship with many of them; adversity brought us together.”*
[IDI2]

### 3.3. THEME 3: New Optics of Critical Care and the Nursing Profession

This theme describes participants’ perceptions of the profession after the acute phase of the pandemic and how it has affected their motivation towards the profession. Although they have felt undervalued by healthcare institutions, for most participants, working in the ICU during the peak of COVID-19 increased their professional and personal motivation. Facing the pandemic has made the nurses become more aware of their profession and has made them reflect on the role of nursing in society, which they had not been confronted with before.


*“It was motivating to see what we are capable of as a profession when the situation demands it, it has also helped boost my professional development. I have learned a lot and believe I have improved as a nurse.”*
[IDI14]

Discovering a passion for critical care nursing.

All participants showed an increase in professional motivation and a sense of self-improvement. Some participants new to the ICU perceived the opportunity to work in this unit as a daily challenge, and since then, have reported feeling like better nurses and more motivated professionally.


*“There are days when you wake up without any motivation, since working as a waitress you would live with less anxiety and less responsibility than as a nurse (...) You have to give yourself the motivation, knowing that, thanks to your work, you can save lives and that, with the resources you are offered, we can make miracles happen.”*
[IDI1]

Despite the difficulties and stress experienced, the attachment to critical patient care has grown in these nurses. In the ICU, they realise how much they can contribute to the profession and therefore, they want to stay in it.


*“I wanted to die when I started, and now I love it. Now I can’t imagine working in any other unit other than the ICU. This is the true essence of being a healthcare professional, saving lives and learning new things, day in and day out, from all the professionals around me.”*
[DI21]


*“My motivation is at its highest. It has always been clear to me that the only important thing in life is health … and I have seen that in the ICU during the pandemic.”*
[IDI15]

Reflections on the nursing profession.

After living through the acute phase of the COVID-19 pandemic, the participants sense a general undervaluation of the profession by health institutions. They emphasise that nurses have played a crucial role throughout the process, but that the profession has been neglected in the process. They feel that their profession has been undervalued and underpaid, with little recognition of the responsibility and risk it involves.


*“Nursing is associated with having a calling. Because you feel a calling, you become a volunteer to work overtime, to not enjoy your days off, to give up vacations, to work holidays and weekends... to give up your life for someone else.”*
[IDI17]


*“We health professionals weren’t cared for then, nor are we cared for now. We face the misery of society, pain, illness and suffering on a daily basis. We deserve better equipment, more staff, better pay and rest, for the work we do.”*
[IDI5]

## 4. Discussion

The aim of our study was to describe and understand ICU nurses’ perceptions of critical patient care in the acute phase of the COVID-19 pandemic. The onset of the COVID-19 epidemic was characterised by a lack of knowledge, lack of information and a constant increase in virus incidence rates [28]. According to other studies [35], the initial lack of control over the situation generated anxiety, fear and pessimism in healthcare professionals, who perceived their workplace as a battlefield against the virus. Exposure to a potentially harmful disease generated psychological distress and depression [26,28], and nurses faced difficulties and negative experiences in caring for their patients. Healthcare providers are on the frontline in response to an epidemic outbreak, exposed to COVID-19 infection during long shifts, with fatigue and exhaustion [1]. Regarding patient care, some participants were satisfied, while others perceived shortcomings. Our participants felt that basic care worsened due to the increased burden of caring for patients [36], understaffing [37,38], and lack of time in emergency situations [39,40]. While hospitals were adjusting staffing to meet the demands of the pandemic, nurses were in direct physical contact with ill patients, putting themselves and their families at risk [6]. This created ethical conflicts for nurses, such as the concern and fear of bringing the virus into their homes and exposing their families to the virus [41,42].

The humanisation of care was thought to compensate for shortcomings in patient care [43]. Nurses showed concern for patients, and accompanied them in coping with illness, delirium [44] or end of life [45]. Restrictions during the pandemic prohibited family visits and company for ICU patients. According to other studies, participants showed greater sensitivity to the suffering and loneliness of patients at their time of death [46] and felt that the dignity of patients and families could have been respected more [47]. Alternatives such as virtual visits had good results, but this innovation was not always possible to use [48,49].

The COVID-19 pandemic modified hospital care practices. The situation overwhelmed veteran nurses, who were suddenly obligated to supervise, guide and support new nurses [50]. In their ICU immersion phase, new nurses assumed patient care duties, but felt powerless because they felt they were a burden to their peers [51,52]. Studies show that not all veteran practitioners are trained to provide supervision [50] and that the new nurses had difficulty integrating into the team [52]. However, our participants noted a sense of teamwork, creating an atmosphere of collaboration between veterans and novices [6]. Working together in an adverse situation strengthened the socialisation among professionals and their connection as a team. Senior nurses were aware of the problems affecting new nurses [53]. Our results support the existence of safety deficiencies due to nurses working long shifts without breaks, caregiving overload and the use of personal protective equipment (PPE) [26,27,35,51,52]. Working in the ICU is challenging for nurses because it is a highly technological and dynamic environment [53]. The experiences of our participants are similar to those at other ICUs in other geographic areas. The nurses felt a lack of experience, a fear of contagion, lack of community support, moral distress and lack of training and competencies [54,55]. Nurses who cared for patients with COVID-19 in ICUs faced a heavy workload, clinical preparedness problems, and noted a lack of available protocols [56]. Although they felt qualified, challenges for the future emerged. The pandemic highlighted planning errors and the need for policies to support nurse specialists in critical care, increase the number of nurses on the teams and protect the public health system [57]. Some novice nurses felt overwhelmed in the early stages, but over time, motivation emerged through finding meaning in their profession [58]. The pandemic brought about a feeling of belonging to the team, professional growth, and reinforcement of the value of nurses as members of the healthcare team [26,35]. However, participants denounce that the profession has not been cared for by society and institutions. According to other studies, participants contemplated leaving the profession after the pandemic [59]. Factors such as stigma, being seen as vectors of contagion, together with the feeling of undervaluation of the profession and neglect by the administration, were most frequently mentioned [24,28].

### Limitations

Nurses with strong feelings about their ICU experiences may have chosen to participate to a greater extent. The principal researcher knows some participants of the sample, which could enhance positive or negative feelings about experiences during the pandemic. The timing of data collection may have impacted the results, as nurses might have forgotten negative or less impactful experiences. Our results should be viewed from the perspective of a medium-sized Level III ICU, although many findings are consistent with other studies. Our data include a single ICU, and we did not collect data from other ICUs during the COVID-19 pandemic: therefore, generalisation of the results is very limited. The study has important implications. Further qualitative research should explore the experiences of healthcare providers in ICUs with different levels of care. The perspective of nursing managers on patient safety and working conditions of ICU nurses during the pandemic should be studied. Social recognition of the role of ICU nurses during the COVID-19 pandemic has not been sufficiently studied; therefore, further studies are needed.

## 5. Conclusions

The COVID-19 pandemic was perceived with harshness by the nurses, as the feeling was like going into a war where everyone was enlisted. The lack of existing knowledge about the virus generated confusion, fear, anxiety and concern about infecting family members and relatives. Although they perceived a lack of humanisation, such as in end-of-life care, they felt that they provided quality care in accordance with the personnel and resources available. Intensive learning experiences shaped their clinical practice; the harsh conditions of isolation, stress, overload and exhaustion marked this stage. While nurses perceived that protective material and resources were available, there was a lack of knowledge of how they functioned, which may have compromised the safety of the professionals. Nurses perceived the ICU as a safe place for the patient, and professionals always prioritised the patient above their personal safety. These experiences generated a strong feeling among nurses of belonging, building increased professional and personal bonds with other team members. Most of them feel the profession receives low recognition, but on a professional level, have gained special motivation for critical care nursing in particular.

## Figures and Tables

**Table 1 healthcare-12-00162-t001:** Sociodemographic data.

Participant	Age	Sex	Marital Status	Children	Work Experience	Experience (ICU)
IDI1	25	Female	Single	-	2 years	1.7 years
IDI2	31	Male	Single	-	2 years	3 years
IDI3	39	Male	Single	-	17 years	3 years
IDI4	32	Female	Single	-	7 years	2 years
IDI5	40	Female	Married	2	18 years	4 years
IDI6	41	Female	Married	2	4 years	1.3 years
IDI7	25	Female	Single	-	4 years	1.1 years
IDI8	38	Female	Divorced	1	15 years	5 years
IDI9	29	Male	Single	-	6 years	3 years
IDI10	33	Female	Single	-	8 years	2 years
IDI11	43	Female	Married	2	19 years	7 years
IDI12	36	Female	Single	-	10 years	3 years
IDI13	27	Male	Single	-	4 years	1 year
IDI14	26	Female	Single	-	3 years	2 years
IDI15	42	Female	Married	3	18 years	16 years
IDI16	37	Female	Married	2	14 years	7 years
IDI17	24	Female	Single	-	3 years	2 years
IDI18	54	Male	Married	1	21 years	13 years
IDI19	36	Female	Married	3	12 years	6 years
IDI20	26	Female	Single	-	3 years	0.8 years
IDI21	31	Male	Single	-	8 years	5 years

IDI = In-depth Interview. ICU = Intensive Care Unit.

**Table 2 healthcare-12-00162-t002:** Development of the interview.

Phase	Themes	Content/Questions
Introduction	Motive	To understand how ICU nurses experience their professional work after the acute phase of the COVID-19 pandemic.
	Objective	Describe nurses’ perceptions.
	Ethics	Information is provided on the voluntary nature of participation in the study, confidentiality of the interviews and their purpose, with the possibility of leaving the study at any time.
Beginning	Introductory question	How did you perceive the onset of COVID-19 in the ICU?
	Semi-structured interview	How did you perceive its influence on patient safety? How did you perceive that it affected the safety of professionals? How did it affect the performance of the team? Is there anything that has not been discussed that you would like to add?

ICU = Intensive Care Unit.

**Table 3 healthcare-12-00162-t003:** Themes, subthemes and units of meaning.

Themes	Subthemes	Units of Meaning
COVID-19 in ICUs: nurses on the frontline.	Nurses on the battlefield: in the trenches	Receiving the call to the front. Ignorance of the disease. Anxiety of having to work. Fear of infecting themselves and their families.
	Questioning the quality of care	Lack of knowledge reduces quality of care. A race against the clock. Excess of occupancy and loss of privacy.
	Humanising care and providing company: death in isolation	Lack of companionship and psychosocial care. Concern for dignity at the end of life. Dying alone, unable to say goodbye.
United against adversity: teamwork.	Observing experienced nurses and learning by doing	Learning by leaps and bounds. Patient care and peer teaching. Lack of experience can compromise care.
	When safety starts to falter	Working in isolation. Lack of knowledge on the use of PPE. Fear of outbreaks in the ICU. Maintaining patient safety. Lack of safety in your profession. Insufficient material resources.
	Coming together to withstand the pandemic	Stressful job. Nervousness and tension. When companionship is lacking. Feeling of union and mutual help.
New optics for intensive care and the nursing profession.	Discovering a passion for critical care nursing	Increased motivation. Professional growth. Capacity for adaptation and resilience.
	Reflecting on the nursing profession	Undervalued nurses. A calling or volunteering. Understanding how others see us. Thinking about my profession.

ICU = Intensive Care Unit. PPE = Personal Protection Equipment.

## Data Availability

Dataset available on request from the authors.

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
