# Peer review of "“Who Takes Care of Carers?”: Experiences of Intensive Care Unit Nurses in the Acute Phase of the COVID-19 Pandemic"

_healthcare, 2024, doi:10.3390/healthcare12020162_

Round 1
Reviewer 1 Report
Comments and Suggestions for Authors
1. Brief Summary and General Comments
Dear authors,
First, I would like to congratulate you on your interesting qualitative study about experiences of nurses working in hospital ICUs during the COVID-19 pandemic. The results of the in-depths interviews with 21 nurses describe how they felt under these exceptional circumstances. The topic is of relevance for Healthcare as it shows the enormous burden nurses had to carry when treating infected patients. You have done a great job. During revision you may consider the following:
2. Title and Abstract
- Please consider writing Intensive Care Unit instead of using the abbreviation (ICU) in the title for clarity.
- Was your objective to describe the experiences of nurses during the outbreak or the acute phase of the COVID-19 pandemic? I recommend being consistent with the wording of your aim (Abstract vs Introduction).
3. Introduction
- Line 41: Please write COVID-19 in capital letters.
- Could you please elaborate more on the theoretical framework applied for your research endeavour?
- Given the qualitative research design, I would kindly advise you to add the research questions to the purpose of the study.
4. Methods
- Line 97: Please add the name and the location (city) of the university where the study was conducted.
- Table 1: Please add the abbreviation ICU to legend of your table explaining IDI and be consistent with the unit used for the ICU experience (IDI76 and IDI7 1.3 / 1.1 months vs IDI20 0.8 years). Two participants had an ICU experience of 1.3 months -> Does this not contradict your inclusion criterion saying that participants needed “at least six months of ICU experience”?
- 2.3 Data collection: How were the data from the in-depth interviews recorded?
- 2.5 Rigour: How was the triangulation done? Please elaborate a bit more on it.
- Please add a reflexivity paragraph to this section (Olmos-Vega et al., 2023).
5. Results
- Line 147: Please recheck the percentage calculation for gender distribution (31.5 % female and 28.5 % male). According to Table 1, it should be 71.4 % female and 28.6 % male.
- Table 3: Please add a legend explaining the abbreviations used in the table.
6. Discussion
- Please consider discussing the limited generalizability of results (data obtained from one study site, no other stakeholders involved) as limitation of your study.
- Please add implications for future research on this topic to the final paragraph.
7. References
- Please check the references so that they are aligned with the guide for submission.
Thank you in advance for considering the comments during revision. Good luck to the authors and kind regards!
References
Olmos-Vega, F. M., Stalmeijer, R. E., Varpio, L., & Kahlke, R. (2023). A practical guide to reflexivity in qualitative research: AMEE Guide No. 149. Medical Teacher, 45(3), 241–251. https://doi.org/10.1080/0142159X.2022.2057287
Comments on the Quality of English LanguageOverall, the manuscript could benefit from some minor editing of English language.
Author Response
See answer in attached file.

Reviewer 2 Report
Comments and Suggestions for Authors
Please see the attachment.

Minor editing of English language required
Author Response
See answer in attached file

Reviewer 3 Report
Comments and Suggestions for Authors
Dear Editor,
Thank you for the opportunity to review this manuscript. This manuscript was well written. The methods and results are presented clearly and the results are discussed well. I just have some minor suggestions and comments for authors:
1. In Table 2, it is not clear which questions or contents belong to which theme. I suggest authors give horizontal lines to separate which contents are for which themes, like authors did in Table 3.
2. In which language were the interviews conducted? If interviews were not conducted in English, and since the results of the study and its related data were reported in English, in which stage of data analysis did the translation to English take place?
3. There is no information about the interview record. Were the interviews audio-recorded? If yes, please add information about any records that were taken during the interviews.
Author Response
See answer in attached file
